# Alpha-Lipoic Acid Attenuates Apoptosis and Ferroptosis in Cisplatin-Induced Ototoxicity via the Reduction of Intracellular Lipid Droplets

**DOI:** 10.3390/ijms231810981

**Published:** 2022-09-19

**Authors:** Sam Cho, Seok Jin Hong, Sung Hun Kang, YongKeun Park, Sung Kyun Kim

**Affiliations:** 1Department of Otorhinolaryngology-Head & Neck Surgery, Hallym University College of Medicine, Dongtan 18450, Korea; 2Department of Biomedical Sciences, Hallym University College of Medicine, Chuncheon 24252, Korea; 3Department of Physics, Korea Advanced Institute of Science and Technology (KAIST), Daejeon 34141, Korea; 4Tomocube, Inc., Daejeon 34141, Korea; 5Laboratory of Brain & Cognitive Sciences for Convergence Medicine, Hallym University College of Medicine, Anyang 14068, Korea

**Keywords:** ferroptosis, ototoxicity, holotomography, lipid droplet, α-lipoic acid

## Abstract

Alpha-lipoic acid (α-LA) is a potent antioxidant that can prevent apoptosis associated with cisplatin-induced ototoxicity through ROS. Ferroptosis is defined as an iron-dependent cell death pathway that has recently been highlighted and is associated with the accumulation of intracellular lipid droplets (LDs) due to an inflammatory process. Herein, we investigated the impact of α-LA on ferroptosis and analyzed the characteristics of LDs in auditory hair cells treated with cisplatin using high-resolution 3D quantitative-phase imaging with reconstruction of the refractive index (RI) distribution. HEI-OC1 cells were treated with 500 μM α-LA for 24 h and then with 15 μM cisplatin for 48 h. With 3D optical diffraction tomography (3D-ODT), the RI values of treated cells were analyzed. Regions with high RI values were considered to be LDs and labelled to measure the count, mass, and volume of LDs. The expression of LC3-B, P62, GPX4, 4-hydroxynonenal (4-HNE), and xCT was evaluated by Western blotting. HEI-OC1 cells damaged by cisplatin showed lipid peroxidation, depletion of xCT, and abnormal accumulation of 4-HNE. Additionally, the count, mass, and volume of LDs increased in the cells. Cells treated with α-LA had inhibited expression of 4-HNE, while the expression of xCT and GPX4 was recovered, which restored LDs to a level that was similar to that in the control group. Our research on LDs with 3D-ODT offers biological evidence of ferroptosis and provides insights on additional approaches for investigating the molecular pathways.

## 1. Introduction

Sensorineural hearing loss, which is mainly caused by ototoxic agents, excessive exposure to noise, genetic disorder, and aging, is a major global burden [1,2,3]. Cisplatin-induced ototoxicity can be caused by various mechanisms of cell death such as apoptosis and autophagy [4,5]. α-Lipoic acid (α-LA), which is a natural antioxidant, is an attractive drug to prevent cisplatin-induced-ototoxicity. It is a key cofactor in mitochondrial metabolism that can chelate iron, copper, and other transition metals by increasing internal cellular glutathione (GSH) levels [6].

Most previous research on the protective effects of α-LA against ototoxicity has focused on apoptosis of HEI-OC1 cells. It was shown that α-LA significantly decreases apoptosis and cell death in cells damaged by cisplatin through regulating MAPKs and cytokines, which inhibits reactive oxygen species (ROS) accumulation in cells and protects mitochondrial function [7,8]. However, the detailed mechanism of the protective effect of α-LA is not fully understood. Cisplatin can induce not only apoptosis but also ferroptosis because it can exhaust GSH and deactivate glutathione peroxidase (GPX) [9]. Indeed, ferroptosis, a recently discovered type of cell death, increases the sensitivity of cisplatin, and inhibition of ferroptosis significantly attenuates cisplatin-induced hair cell damage by protecting mitochondrial function in hair cells [10,11,12,13].

Liproxstatin-1 and ferrostatin-1 are known to suppress ferroptosis. Liproxstatin-1 conserves GPX4 and regulates acyl-CoA synthetase long-chain family member 4 and cyclooxygenase 2. Ferrostatin-1 inhibits lipid peroxidation by para-catalytic methods, creating anti-ferroptotic effects similar to GPX4 [14,15]. α-LA, which can regenerate intrinsic antioxidants in the body including GSH, is also worth investigating as a ferroptosis inhibitor because of its curing mechanism in HEI-OC1 cells damaged by cisplatin.

Ferroptosis is an iron-dependent form of cell death caused by the lethal accumulation of lipid-based ROS when lipid peroxide repair systems are damaged. The accumulation of lipid peroxides is mainly caused by the reduced activity of GPX4, which is a unique intracellular antioxidant enzyme that suppresses lipid peroxidation in the cell membrane [16,17]. In addition, ferroptosis is related to a deficiency in the lipid peroxidation removal ability and has a close association with lipid droplets (LDs) [18,19]. LDs are identified as ubiquitous dynamic organelles that synthesize, store, and supply lipids in the cells of various organisms. LDs accumulate highly when cells are under infectious and inflammatory conditions, and LDs function can be changed by metabolic imbalances and medications [20,21]. There is a clear relationship between the formation of LDs and cell death, but the quantitative changes in LDs in cells during various cell death pathways including ferroptosis are still not defined.

Imaging of LDs in living cells is more suitable for research on biochemical phenomena and has the advantage of avoiding the deformation of LDs structure due to the use of lipophilic organic dyes [22,23]. Thus, 3D optical diffraction tomography (3D-ODT) was used to observe LDs because it allows high-resolution imaging without chemical dyeing of living cells. 3D-ODT, using a Mach–Zehnder interferometric microscope with an integrated digital micromirror device, can accumulate data by capturing 2D-holograms from various angles and illuminations to obtain 3D-phase images [24,25]. Users can reorganize the accumulated refractive index (RI) distributions and label cell organelles with the RI values of the image for quantitative measurement [26].

In this research, we aimed to verify the protective effect of α-LA against ferroptosis mediated by cisplatin-induced ototoxicity and quantitively analyzed the characteristics of intracellular LDs using the 3D-ODT imaging technique.

## 2. Results

### 2.1. Cisplatin Induces Cytotoxicity of HEI-OC1 Cells, While α-LA Protects against Cisplatin-Induced Damage to Auditory Hair Cells

The 15 μM cisplatin group showed approximately 50% of the viability of the control group (Figure 1A). The viability of cells treated with 1000 μM α-LA was significantly lower than 100 μM and 500 μM, but was similar to that of control cells (Figure 1B). Figure 1C indicates that the α-LA/Cis group, which was pretreated with 500 μM α-LA, showed similar results to the control group, whereas the cisplatin group without α-LA treatment showed significantly decreased cell viability. 

### 2.2. HEI-OC1 Cell Damage Related to Cisplatin-Induced Ototoxicity Causes Quantitative Changes in LDs

The 2D isosurfaces from the RI tomography of the cisplatin, α-LA/Cis, and control groups are shown in Figure 2A–C, respectively. The cell membrane and cell nucleus appeared in the RI range of 1.337 to 1.434. To confirm that the high-RI regions corresponded to the LDs of HEI-OC1 cells, LipiDye (#FDV-0010, Tokyo, Japan) was used to visualize LDs in the cytoplasm (Figure 2E). The LDs and high-RI regions (shown in green and red, respectively) showed over 90% correspondence. The count, volume, and mass of LDs were analyzed for cells labeled with each RI value (Figure 2D–G). The LDs count was significantly higher in the cisplatin group than in the control group (*p* < 0.001). On the other hand, the LDs count in the α-LA/Cis group was significantly lower than that in the cisplatin group but higher than that in the control group (*p* < 0.001) (Figure 2H). Figure 2I,J indicate that the volume and the mass of LDs were significantly increased in the cisplatin group compared to the control group (*p* < 0.001). The volume and mass were decreased in the α-LA/Cis group compared to the cisplatin group (*p* < 0.001).

### 2.3. Cisplatin-Induced Ototoxicity Causes Apoptosis in HEI-OC1 Cells via PARP Cleavage through ROS Increases and Caspase-3 Activation

The levels of ROS and the activity of Caspase-3 and poly (ADP-ribose) polymerase (PARP) in cells from the cisplatin, α-LA, and control groups were analyzed. The expression of 2′,7′-dichlorofluorescein (DCF), which is formed by oxidation of DCFH-DA, was increased in the cisplatin group than in the control group (*p* < 0.001) (Figure 3A,B). Similarly, the activity of Caspase-3 was higher in the cisplatin group than in the control and α-LA/Cis groups (*p* < 0.001). Western blotting and band intensity analysis revealed remarkably high levels of cleaved PARP in the cisplatin group (*p* < 0.01). the α-LA/Cis group showed lower ROS and Caspase-3 activity levels that were similar to those in the control group, and the band for cleaved PARP was not found in the α-LA/Cis group.

### 2.4. HEI-OC1 Cells Show Progression toward Ferroptosis Involving Various Cell Death Pathways after Cisplatin Treatment

The levels of apoptosis, autophagy, and ferroptosis in each group were analyzed through immunofluorescence and Western blotting (Figure 4A–J). The cisplatin group showed lower expression of p62 and LC3-I than the control and α-LA/Cis groups but higher expression of LC3-II than that of the control group (*p* < 0.001). The expression levels of xCT and GPX4 were lower in the cisplatin group than in the control and α-LA/Cis groups. Accordingly, 4-HNE expression in the cisplatin group was higher than that in the other groups (*p* < 0.01). The α-LA/Cis group showed higher p62 activity (*p* < 0.01) and LC3-II expression (*p* < 0.001) than those of the control and cisplatin groups. GPX4 expression in the α-LA group was also higher than that in the control group and cisplatin group (*p* < 0.05 and *p* < 0.01, respectively). 4-HNE expression in the α-LA/Cis group was higher than that in the control group (*p* < 0.01) but lower than that in the cisplatin group (*p* < 0.01).

## 3. Discussion

Recently, it was found that ferroptosis, one of the cell death mechanisms, increases the sensitivity against cisplatin-induced ototoxicity by reducing mitochondrial function of hair cells in the inner ear [10,11,12]. Similarly, this phenomenon can be observed in HEI-OC1 cells during the progression of ferroptosis in the contexts of cell damage and inflammation [18,20]. Investigating the LDs in ferroptosis can provide a reason for the accumulation of lipids and oxides, which can be key to analyzing the molecular mechanisms.

In this research, quantitative changes in intracellular LDs were analyzed through intracellular bioimaging with 3D-ODT. HEI-OC1 cells that were damaged by cisplatin underwent various cell death mechanisms including ferroptosis and showed significantly quantitative changes in LDs. Due to the characteristics of the cell structure, most research measures LDs form and viscosity through imaging techniques using fluorescent probes. However, quantitative inspection of LDs in damaged HEI-OC1 cells had not been performed until now. In this research, LDs were labelled and quickly measured based on the RI values from the label-free 3D-ODT without specific fluorescent probes.

LDs have different forms according to the cell type. LDs can exist either in a homogeneous form with a small size or in a dispersed and localized form. In the case of mammals, LDs can exist in the form of a single large lipid droplet [27,28]. LDs are known to act as a sponge to absorb excessive fatty acids during oxidative stress to protect cells from lipid toxicity [29]. According to the result of this research, cisplatin-induced ototoxicity reduces the viability of HEI-OC1 cells and activates caspase-3, which results in high ROS production and cell death. These effects increase the cleavage of PARP and ultimately lead to apoptosis. However, it was found that cells were recovered to the level of normal cells when the HEI-OC1 cells were pre-treated with α-LA. The cisplatin group and α-LA/Cis group both showed high autophagy activity, but differ in the size and density of LDs.

Autophagy can contribute to the inflammatory response of LDs and is related to cell death either positively or negatively. In the cisplatin group [30,31], p62 interacted with LC3, and its level decreased because of decomposition via the production of autophagosomes [32]. Decreases in p62 and LC3-I may be involved in autophagosome-lysosome fusion, which causes dynamic activation of autophagy. However, the accumulation of LC3-II indicates that autophagosomes are not decomposed, and overproduction of autophagosomes can be involved in cell death [33,34]. Cell death through autophagy can be regulated by genetic manipulation and drugs [35]. Autophagy can occur simultaneously with apoptosis and ferroptosis, and this phenomenon shows that autophagy can contribute to the inflammatory response of ferroptotic cell death under specific conditions [36]. For example, excessive activity of autophagy and lysosomes can stimulate ferroptosis by iron accumulation or lipid peroxidation [37,38,39]. Alternatively, erastin and sorafenib, as known inducers of ferroptosis, improve the expression of SQSTM1/p62, which leads to the removal of KEAP1 protein through autophagy, stabilization of NFE2L2 protein, and deactivation of ferroptosis [40,41]. Thus, the result of the cisplatin group having low expression of GPX4 and xCT, which inhibit oxidation damage, and high lipid peroxidation, including quantitative changes in LDs, implies that the damage to HEI-OC1 cells by cisplatin is negatively related to apoptosis and ferroptosis through autophagy, which results in quantitative changes in the LDs count, mass, and volume. The α-LA/Cis group treated with α-LA also showed p62 protein activation and high LC3-I and LC3-II production. This finding implied that autophagosomes, which represent the early stage of autophagy and can positively affect cells with damage and inflammation, were actively produced. Normal autophagy reduces LDs formation in cells and contributes to intercellular homeostasis [42,43]. As a result, cells in the α-LA/Cis group recovered as their cell death progression was suppressed. According to the 3D-ODT RI analysis, LDs in the α-LA/Cis group might be reduced by autophagy. Nevertheless, the biological meaning of the various quantitative changes in LDs by cisplatin is still not clear, and additional research including proteomics and lipidomics of LDs is necessary.

In summary, in this research, the formation and metabolic processes of LDs were analyzed by measuring the 3D-RI values of living cells through 3D-ODT without specific markers. Furthermore, protective effects on HEI-OC1 cells were quickly and precisely analyzed. The existing imaging techniques without labeling have many problems such as a low resolution, difficulty obtaining RI information for samples, a need for high-power laser sources, and long acquisition times [44,45]. However, the 3D-ODT used in this research is practical in that it can quickly and easily distinguish and measure intracellular lipid accumulation [46,47]. The unique LDs measuring technique used in this research can offer biological evidence about ferroptosis of the inner ear and can be an effective approach to additional research on the molecular pathways of ferroptosis.

## 4. Materials and Methods

### 4.1. Materials

Cisplatin (product no. 15663-27-1) and α-lipoic acid (product no. 1077-28-7) were purchased from Sigma Chemical Co. (St. Louis, MO, USA). Ethanol (99.5%) was specifically used as a solvent to α-LA. Cisplatin was dissolved into PBS and stirred until particles were completely dissolved.

### 4.2. Cell Culture

The HEI-OC1 cell line was cultured under permissive conditions at 33 °C in a humidified incubator with 10% CO_2_. The cells were maintained in high-glucose Dulbecco’s modified Eagle’s medium (DMEM; Gibco BRL, Gaithersburg, MD, USA) containing 10% fetal bovine serum (FBS; Gibco BRL) and 50 U/mL gamma interferon without antibiotics [48]. The cell line was developed as an in vitro system to investigate the cell morphological mechanisms involved in ototoxicity and to screen potential ototoxic agents.

### 4.3. Cell Counting Kit-8 (CCK-8) Assay of Cell Viability

Cell viability was evaluated with a CCK-8 assay (Dojindo Laboratories, Kumamoto, Japan; CK04). HEI-OC1 cells were seeded in 48-well plates; each well contained 20,000 cells with 0.5 mL of complete growth medium. The cells were 70% confluent after 24 h. The cells were pretreated with 500 μM α-LA for 24 h before cisplatin exposure. The cells were then incubated with 15 µM cisplatin for 48 h. All cells were then incubated with 10 μL of CCK-8 reagent and 90 μL of serum-free medium at 33 °C for 2 h. The optical density was measured with an Epoch Microplate Spectrophotometer (Bio Tek Instruments, Winooski, VT, USA) at 570 nm.

### 4.4. Fluorescence Microscopy ROS Assay

In brief, cells were washed twice with prewarmed serum-free medium without phenol red and incubated with the fluorescent dye 2′,7′-dichlorofluorescein diacetate (DCFH-DA) at 50 μM for 30 min at 33 °C. After washing with serum-free medium without phenol red, the cells were fixed with 3.7% glutaraldehyde and Hoechst (#33258 Sigma, St. Louis, MO, USA) for 15 min at room temperature in the dark. Fluorescence was then observed with a confocal microscope (ZEISS LSM-800, Jena, Germany).

### 4.5. Measurement of Caspase-3 Activity

The enzymatic activity of caspase-3 was assayed with a caspase-3/CPP32 fluorometric assay kit (BioVision, Milpitas, CA, USA). The cells were harvested and resuspended in 200 μL of lysis buffer. The protein concentration of each sample was quantified with bovine serum albumin (BSA; #23209 Thermo Fisher Scientific, Waltham, MA, USA). The protein was incubated with the fluorometric substrate DEVD-AFC at 50 mM for 2 h at 33 °C. Then, it was measured with a FlexStation 3 Multi-Mode Microplate Reader (Molecular Devices, San Jose, CA, USA) at a 400 nm excitation wavelength and a 505 nm emission wavelength.

### 4.6. Immunofluorescence Staining

HEI-OC1 cells were cultured in 24-well dishes with DMEM plus 10% FBS. Briefly, the cells were fixed with 4% paraformaldehyde for an hour and then permeabilized with 0.25% Triton X-100 for 30 min. After that, the cells were blocked with 1% BSA in PBS for an hour and then incubated overnight with an LC3-B primary antibody (#2775, 1:1000; Cell Signaling Technology, Danvers, MA, USA) and a 4-hydroxynonenal primary antibody (#BS-6313R, 1:200; Bioss, Woburn, MA, USA) at 4 °C. After three washes with PBS, the cells were stained with an Alexa Fluor-555 anti-rabbit antibody (#4413, 1:2000; Cell Signaling Technology, Danvers, MA, USA), Hoechst (#33258, Sigma, St. Louis, MO, USA), and LipiDye (#FDV-0010, Tokyo, Japan) for 10 min in the dark. The specimens were then observed under a laser scanning confocal microscope (ZEISS LSM-800, Jena, Germany).

### 4.7. Holotomography

HEI-OC1 cells were cultured in 48-well dishes with DMEM plus 10% FBS. The cells were pretreated with 500 μM α-LA for 24 h and then treated with 15 μM cisplatin for 48 h. The treated cells were subcultured for 24 h in a central glass-bottom TomoDish (Tomocube, 901002-02), washed with prewarmed serum-free medium without phenol red, incubated and then stained with LipiDye (#FDV-0010 Tokyo, Japan) for at least 2 h at 33 °C. The LypiDye was removed, and a coverslip was placed on top of the cells before tomography. Twenty independent experiments were performed. The RIs of 600 cells from the control, cisplatin, and α-LA/Cis groups were labeled and compared to confirm the differences in LDs in HEI-OC1 cells between the groups. High-RI regions were regarded as LDs [49], and LipiDye was used to visualize LDs in the cytoplasm for cross-checking. 3D-ODT was performed and LipiDye fluorescence images of live cells were obtained with a Mach–Zehnder interferometric microscope that combines holography and fluorescence images into a single image. As shown in Figure 5, the 3D-ODT device reorganized the RI distributions in 2D holograms captured from various angles of illumination.

### 4.8. Western Blot Analysis

HEI-OC1 cells were washed with PBS, and total protein was extracted from the cells using CETi Lysis Buffer with inhibitors (#TLP-121CETi Translab, Daejeon, Korea). The protein concentration was measured with BSA (#23209 Thermo Fisher Scientific, Waltham, MA, USA), and a total of 20 μg of protein was denatured at 95 °C and separated by 10% or 12% SDS–PAGE. The proteins were then transferred to PVDF membranes by electrophoresis and blocked with 5% milk-containing Tris-buffered saline and 0.1% Tween (TBST) buffer at room temperature (RT). The membranes were incubated with primary polyclonal antibodies diluted 1:2000 at 4 °C overnight. After washing with TBST, the membranes were incubated with peroxidase-conjugated secondary antibodies diluted 1:3000 (#7074 Cell Signaling Technology, Danvers, MA, USA), and the proteins were visualized with the chemiluminescence solution in SuperSignal West Dura Extended Duration Substrate (#34075 Thermo Fisher Scientific, Waltham, MA, USA). To calculate the band density, ImageJ software was used (Broken Symmetry Software, United States). The primary antibodies included anti-PARP (product no. #9542), anti-LC3-I, II (product no. #2775), anti-p62 (product no. #5114), and anti-beta-actin (product no. #4967), which were purchased from Cell Signaling Technology (Danvers, MA, USA), and anti-4-HNE (BS-6313R), which was purchased from Bioss (Woburn, MA, USA).

### 4.9. Statistical Analysis

Statistical analysis was performed using SPSS (IBM Inc., NY, USA). The data were presented as the mean ± standard deviation (SD). Paired t tests were used for paired data, while multiple datasets were analyzed by one-way analysis of variance (ANOVA). Tukey’s honest significant difference test was conducted to determine which group means differed. *p* < 0.05 was considered to indicate statistical significance.

## 5. Conclusions

HEI-OC1 cell damage induced by cisplatin ototoxicity caused autophagy-dependent ferroptosis including apoptosis and quantitatively increased the numbers, masses, and volumes of LDs. α-LA mitigated ferroptosis as well as apoptosis against cisplatin-induced ototoxicity, suggesting that changes in the characters of intracellular LDs observed under holotomography may be associated with ferroptosis. These results show that a protective mechanism of α-LA recovers parameters of LDs to normal cell’s level through control of oxidation stress and activation of normal autophagy in ototoxicity. This study demonstrates that quantitative analysis of LDs with a 3D-ODT device can be used to quickly monitor and distinguish the progress of ferroptosis.

## Figures and Tables

**Figure 1 ijms-23-10981-f001:**
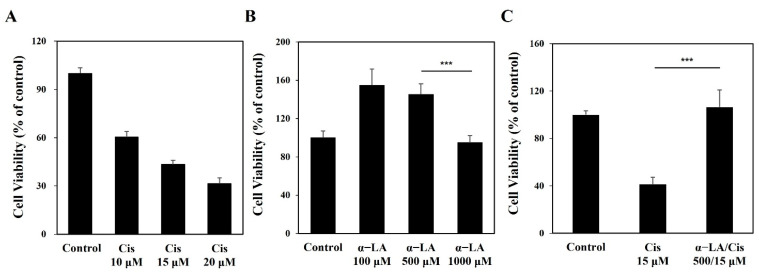
Viability of HEI-OC1 cells treated with cisplatin and α-LA. (**A**) Cisplatin-induced cytotoxicity increased in a dose-dependent manner. (**B**) The toxicity test of α-LA toward HEI-OC1 cells showed a significant decrease at the concentration of 1000 μM. (**C**) α-LA showed protective effects against cisplatin-induced damage in HEI-OC1 cells. (*** *p* < 0.001, determined using an independent *t* test). The group treated with cisplatin alone is abbreviated as Cis, while the group with α-LA pretreatment is abbreviated as α-LA/Cis.

**Figure 2 ijms-23-10981-f002:**
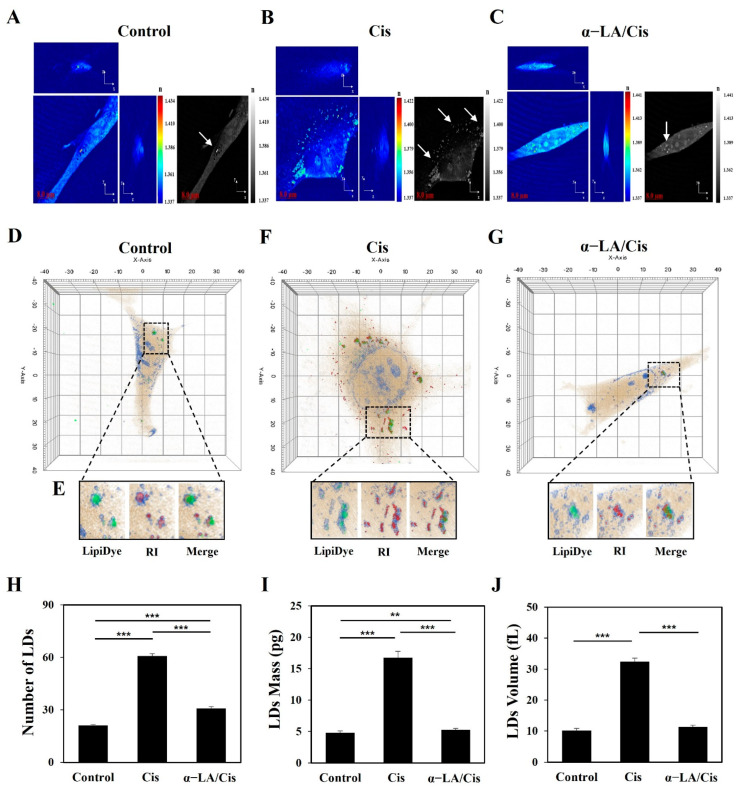
The measurement of LDs through 3D-ODT using the RI values of HEI-OC1 cells. (**A**–**C**) 2D isosurface images for the control group, cisplatin group, and α-LA/Cis group, respectively. The white arrows indicate LDs (high-RI regions), and the colored bars indicate the 2D-rendered RI range of 1.3 to 1.4 with an 8 μm scale bar. (**D**–**G**) Cell images labeled by RI analysis. The LDs regions in cells are marked with black dotted lines. The region of LDs under the LDs stain in (**E**) is marked in green, and the RI is marked in red. (**H**–**J**) LDs numbers, masses, and volumes, respectively, measured through RI analysis of 3D images showed high levels in the cisplatin group, while the levels are recovered in the α-LA/Cis group. In 20 independent experiments, 600 cells from each group were compared, and the results are presented as the means ± SDs (** *p* < 0.01, and *** *p* < 0.001, determined using one-way ANOVA).

**Figure 3 ijms-23-10981-f003:**
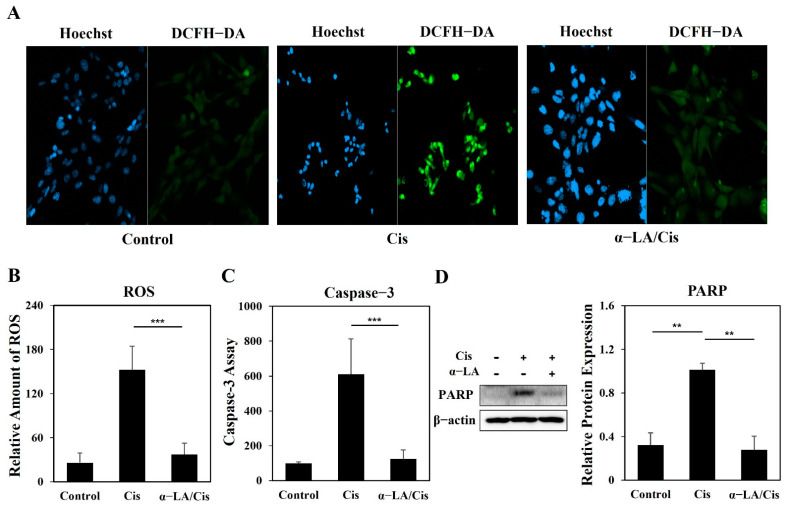
Analysis of expression of intracellular ROS, caspase-3 and PARP in HEI-OC1 cells damaged by cisplatin-induced ototoxicity and protective effect of α-LA. (**A**) Imaging of ROS in each group with DCFH-DA staining. The left panel shows cell nuclei stained with Hoechst 33342, and the right panel shows an enlarged DCFH-DA image (×200). (**B**) The level of ROS for each group were analyzed, and the results are presented in a graph. (**C**) Caspase-3 activity was analyzed with a fluorometric kit using a substrate. (**D**) Western blot analysis was performed with antibodies targeting PARP. The level of ROS, Caspase-3, PARP in HEI-OC1 cells damaged by cisplatin was recovered after α-LA treatment. The data are presented as the means ± SDs of triplicate determinations from three independent experiments (** *p* < 0.01, and *** *p* < 0.001, determined using one-way ANOVA).

**Figure 4 ijms-23-10981-f004:**
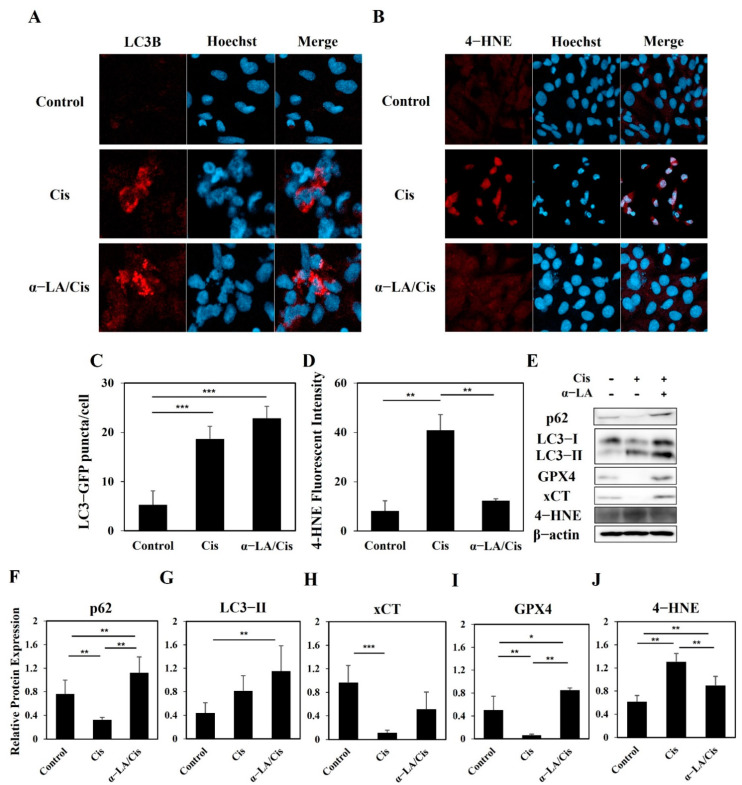
Fluorescence microscopic images and changes in cell death markers of HEI-OC1 cells caused by cisplatin-induced ototoxicity. (**A**) Immunofluorescence staining for LC3-B (red) (×400). (**B**) Immunofluorescence staining with 4-HNE (red) (×400). (**C**,**D**) Both the cisplatin and α-LA/Cis groups showed high fluorescence intensities of LC3-II, while only the α-LA/Cis group showed low intensity of 4-HNE. (**E**–**J**) According to the protein expression result, the α-LA/Cis group had significantly recovered expression of GPX4, which leads to low expression of 4-HNE. The data are presented as the means ± SDs of triplicate determinations from three independent experiments (* *p* < 0.05, ** *p* < 0.01, and *** *p* < 0.001, determined using one-way ANOVA).

**Figure 5 ijms-23-10981-f005:**
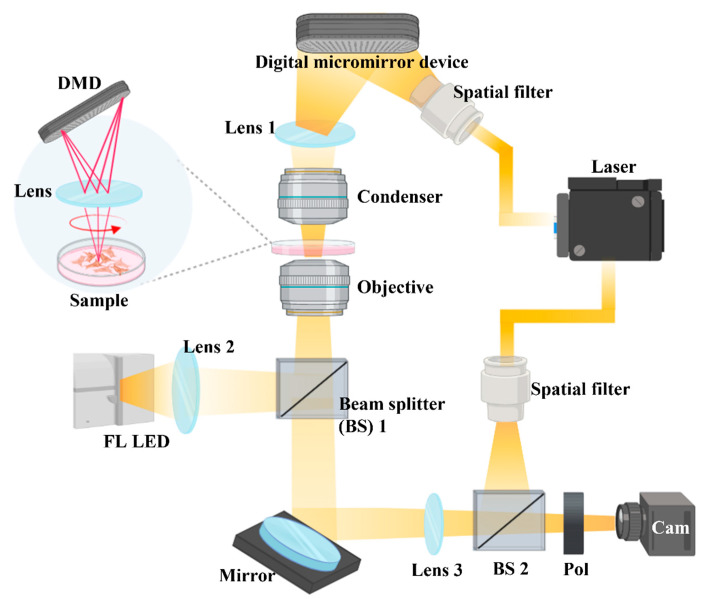
3D-ODT device. A 3D-ODT device was used to capture 3D images of LDs in the cisplatin group and α-LA group. With a Mach–Zehnder interferometric microscope with an integrated digital micromirror device, 2D holograms were captured from various angles of illumination, and the RI distributions of these holograms were reorganized to obtain 3D-phase images.

## Data Availability

Not applicable.

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
