# Peer review of "Alpha-Lipoic Acid Attenuates Apoptosis and Ferroptosis in Cisplatin-Induced Ototoxicity via the Reduction of Intracellular Lipid Droplets"

_ijms, 2022, doi:10.3390/ijms231810981_

Round 1
Reviewer 1 Report
Lipoic acid as an essential cofactor for mitochondrial metabolism plays an essential role in stabilizing and regulating mitochondrial multienzyme complexes. Moreover, it is responsible for cellular growth and mitochondrial activity as well as necessary for the coordination of energy metabolism. In the latest generation of antioxidants, lipoic acid can alleviate the hydroxyl radical, hypochlorous acid, singlet oxygen, and peroxy radicals. Besides, they can chelate iron, copper, and other transition metals by increasing intracellular reduced glutathione (GSH) levels. Therefore it is worth investigating the therapeutic effect of lipoic acid in cisplatin induced processed of ferroptosis, especially using the method using high-resolution 3D quantitative-phase imaging.
My only complaint is the conclusion in the introduction that does not match the title. The effect of lipoic acid is not seen. The conclusion refers only to the advantage of the 3D-ODT method in the detection of LDs. A similar thing happens in the final conclusion. The authors shyly present the contribution of lipoic acid in relation to the advantage of the 3D-ODT method for detection as if they wrote a methodological paper and then changed their minds.
Author Response
The revisions made after carefully considering the comments of the reviewer#1 as follows. (Note: reviewer comments are in italics; our responses are in light blue.)
Reviewer #1:
Lipoic acid as an essential cofactor for mitochondrial metabolism plays an essential role in stabilizing and regulating mitochondrial multienzyme complexes. Moreover, it is responsible for cellular growth and mitochondrial activity as well as necessary for the coordination of energy metabolism. In the latest generation of antioxidants, lipoic acid can alleviate the hydroxyl radical, hypochlorous acid, singlet oxygen, and peroxy radicals. Besides, they can chelate iron, copper, and other transition metals by increasing intracellular reduced glutathione (GSH) levels. Therefore, it is worth investigating the therapeutic effect of lipoic acid in cisplatin induced processed of ferroptosis, especially using the method using high-resolution 3D quantitative-phase imaging.
My only complaint is the conclusion in the introduction that does not match the title. The effect of lipoic acid is not seen. The conclusion refers only to the advantage of the 3D-ODT method in the detection of LDs. A similar thing happens in the final conclusion. The authors shyly present the contribution of lipoic acid in relation to the advantage of the 3D-ODT method for detection as if they wrote a methodological paper and then changed their minds.
: Thank you for your valuable comments. We added the effect of a-lipoic acid to the conclusion of the manuscript. We added this information as follows.
Page 9: “α-LA mitigated ferroptosis as well as apoptosis against cisplatin-induced ototoxicity, suggesting that changes in the characters of intracellular LDs observed under holotomography may be associated with ferroptosis. These results show that a protective mechanism of α-LA recovers parameters of LD to normal cell's level through control of oxidation stress and activation of normal autophagy in ototoxicity.”
Reviewer 2 Report
Review of the paper entitled “Alpha-lipoic acid attenuates ferroptosis in cisplatin-induced ototoxicity via the reduction of intracellular lipid droplets” by Sam Cho, Seok Jin Hong, Sung Hun Kang, YongKeun Park, and Sung Kyun Kim.
Lipoic acid (LA) is a very interesting molecule. It plays important roles in a wide range of physiological and pathological conditions, as well as in therapy. The aim of the Authors was to verify the protective effect of LA against ferroptosis mediated by cisplatin (Cis)-induced ototoxicity. The research was carried out on the House Ear Institute-Organ of Corti 1 (HEI-OC1) cell line. It is one of the few mouse auditory cell lines currently available for research goals. Ferroptosis is a type of programmed cell death that is usually accompanied by a large amount of iron accumulation and lipid peroxidation during the cell death process. Moreover, ferroptosis has a close association with the accumulation of intracellular lipid droplets (LDs) due to an inflammatory process. The most toxic product of lipid peroxidation is 4-hydroxy-2,3-trans-nonenal (HNE). The Authors used high-resolution 3D quantitative-phase imaging with reconstruction of the refractive index (RI) distribution. The HEI-OC1 cells were treated with LA for 24 hours and then with Cis for 48 hours. With 3D optical diffraction tomography (3D-ODT), the RI values of treated cells were analyzed. The results obtained by the Authors indicated that the α-LA/Cis group showed low fluorescent intensity of HNE. Moreover, the Authors observed that LDs in the α-LA/Cis group might be reduced by autophagy. In addition, the Authors emphasize that the use of the 3D-ODT technique offers biological evidence of ferroptosis and provides insights on additional approaches for investigating the molecular pathways. The paper is interesting and well written.
My comments
The Materials and Methods section should contain information on the preparation of lipoic acid (LA) and cisplatin (Cis) solutions. It is known that both of these compounds are quite difficult to dissolve in aqueous solutions. So, what solvents did the authors use to dissolve LA and Cis? Have the effects of the solvents been tested on HEI-OC1 cells?

Author Response
The revisions made after carefully considering the comments of the reviewer#1 as follows. (Note: reviewer comments are in italics; our responses are in light blue.)
Reviewer #2:
The Materials and Methods section should contain information on the preparation of lipoic acid (LA) and cisplatin (Cis) solutions. It is known that both of these compounds are quite difficult to dissolve in aqueous solutions. So, what solvents did the authors use to dissolve LA and Cis? Have the effects of the solvents been tested on HEI-OC1 cells?
: Thank you for valuable comments. We added the information about preparation of lipoic acid and cisplatin solutions. We added this information as follows.
Page 7: “Ethanol (99.5%) is specifically used as a solvent to α-LA. Cisplatin is dissolved to PBS and it is stirred until particles is completely dissolved.”
According to recommendations of the manufacturer, 0.253g of cisplatin is soluble to 100g of water. When we prepare cisplatin, 5mM of stock is used (0.0015g/ml per 1 stock). It is stirred until fully dissolved and used for 3 months while refrigerated.
α-Lipoic acid is not easily dissolved to water. Thus, ethanol is used as a solvent, and we initially prepare 100mM/mL (0.02g/mL) of stocks and use it while refrigerating stocks (Recommendations of the manufacturer prefer 0.05g/mL of ethanol). When fully dissolved 100mM lipoic acid stock is used to the experiment, it is diluted to 1mM/mL and placed to culturing medium (about 1% of ethanol contained). About 1% of ethanol content in culturing medium is the level that cells can endure [1-3].
References:
- Nguyen, S. T.; Nguyen, H. T.-L.; Truong, K. D., Comparative cytotoxic effects of methanol, ethanol and DMSO on human cancer cell lines. Biomedical Research and Therapy 2020, 7, (7), 3855-3859.
- Wakabayashi, I.; Negoro, M., Mechanism of inhibitory action of ethanol on inducible nitric oxide synthesis in macrophages. Naunyn-Schmied eberg's archives of pharmacology 2002, 366, (4), 299-306.
- Forman, S.; Káˇs, J.; Fini, F.; Steinberg, M.; Ruml, T. s., The effect of different solvents on the ATP/ADP content and growth properties of HeLa cells. Journal of biochemical and molecular toxicology 1999, 13, (1), 11-15.